# An In-Situ Assessment of Wood-in-Service Using Microwave Technologies, with a Focus on Assessing Hardwood Power Poles

**DOI:** 10.3390/insects11090568

**Published:** 2020-08-25

**Authors:** Graham Brodie, Deepan Babu Thanigasalam, Peter Farrell, Allison Kealy, John R. J. French, Berhan Ahmed (Shiday)

**Affiliations:** 1Faculty of Veterinary and Agricultural Sciences, University of Melbourne, Parkville, VIC 3010, Australia; tdbabu@yahoo.com.au (D.B.T.); feetapparel@gmail.com (P.F.); b.ahmed@unimelb.edu.au (B.A.); 2Geospatial Sciences, Royal Melbourne Institute of Technology University, Melbourne, VIC 3000, Australia; allison.kealy@rmit.edu.au; 3Faculty of Science, Health, Education and Engineering, University of the Sunshine Coast, Maroochydore, DC, QLD 4558, Australia; ecospan@internode.on.net

**Keywords:** wood, wooden, poles, microwave, radar, termites, decay

## Abstract

**Simple Summary:**

Both termites and decay fungi use wood as a food source. Termites and decay cause major damage and inflict significant costs on human societies around the world. Currently, power pole testing techniques either involve a subjective evaluation, based on the sound that the pole makes when it is hit by a heavy hammer, or observing wood shaving from holes that are drilled into the core of the pole itself. Drilling is destructive and compromises any protection from wood preservative treatments, allowing termites or decay fungi to enter the pole. This project developed a simple, objective measurement technique, which is based on the transmission of very low power microwave energy through the wood. Two versions of the system were developed, and both could easily distinguish between sound hardwood poles and those which were compromised by decay and termite attack, without compromising the pole’s integrity in any way. These will be of enormous benefit to the power utility industry.

**Abstract:**

Wooden power poles and their ongoing inspection represent a significant investment for most electrical power utilities. This study explored the potential for using microwave fields to non-invasively assess the state of hardwood power poles in a field experiment. Two strategies were assessed: 2.4 GHz microwave field transmission through the pole; and mutual coupling between antennae using a 10.525 GHz radar module applied to the surface of the pole. Both systems distinguished between sound hardwood poles and those which were compromised by decay and subterranean termite attack and infestation.

## 1. Introduction

The wooden power pole inventory in Australia was estimated to be in excess of five million in 2004 [1]. In the European Union, the population of wooden power poles has been estimated to be approximately one for each two inhabitants [2]. The ongoing inspection and maintenance of this inventory represents a significant investment in time and money for most electrical utilities, with many of the testing techniques involving some level of destructive testing.

Several strategies are being explored to quickly and cheaply detect bio-deterioration and subterranean termite (‘termite’) attack and infestation in timber-in-service. Microwave technologies lend themselves to this task because, unlike other forms of electromagnetic radiation, microwaves have quite long waves allowing deep penetration of dielectric materials, like seasoned timber. Microwave fields also interact strongly with moisture [3,4,5,6,7], providing an excellent opportunity to identify termite and decay activity in an otherwise dry hardwood environment [8], such as power poles.

A number of microwave based biosecurity applications have been explored recently, including the detection and control of insects in bulk materials such as grains and wood [9,10,11]. When microwaves are transmitted through wood, the wave will be partially reflected, attenuated and delayed compared to a wave travelling through free space (Figure 1). The extent of these three effects depends on the bulk dielectric properties of the material. The combination of wave attenuation and delay provides useful information about the material through which the wave passed. An x-ray image is a good example of the information that can be derived from wave attenuation and delay.

The Termatrac^®^ system (Termatrac, Newport Beach, CA, USA) is a commercial example of using microwave energy to detect termite activity in timber. It is a radar based system, which responds to movement [12]. The maximum reliable wood depth for detecting termites was 5 cm [13], which is limiting for evaluation of larger wooden structures, such as power poles. Evaluation of the Termatrac^®^ system revealed that operator skill was critical in establishing its efficacy in termite detection. For example, Zahid, et al. [12] found that one of their test operators appeared to be better than the other with a sensitivity of 66% compared to 45% for the second operator. While this could be overcome with training, their study found that the efficacy of the Termatrac^®^ system was lower than several other termite detection systems. Another limitation of the Termatrac^®^ system is that it is expensive and not convenient for permanent installation.

It has also been shown that mutual coupling of microwave fields occurs between antennas in the same geometric plane [14,15,16,17]. The extent of this mutual coupling will also depend on the bulk dielectric properties of the space in front of the antennas’ plane.

### 1.1. Properties of Wood

The dielectric properties of wood at microwave frequencies depend on: the frequency of the microwave fields; the density of the wood; the orientation of the microwave’s electrical field with respect to the wood grain; the uniformity of the wood structure in front of the antennas (i.e., the presence of large voids in the wood will change the bulk dielectric behaviour of the sample); and especially the moisture content of the wood (Figure 2) [7].

At the micro-structural level, wood can be described in terms of longitudinal tracheids, pit pairs, which connect between tracheids, the primary cell walls and the secondary cell walls [19,20]. Hardwoods contain vessel elements [21,22] while softwoods often contain resin channels [21]. Hardwood vessels serve as the pipelines within the trunk, transporting sap within the living tree. They extend through the trunk and branches, providing longitudinal connections along the length of the timber. Resin channels are resin filled cavities in the wood structure. Both vessels and resin channels affect the physical and electromagnetic properties of timber.

The orientation of wood cells profoundly affects all the measurable properties of wood [7]. In particular, the microwave’s electrical field will strongly interact with elements that are aligned in the same plane as this field vector. In the case of wood, the elements that most affect microwave fields will be: the tracheid cell walls found only in softwoods, which are oriented along the grain of the wood; vessel walls, which are a feature of hardwoods only and also run along the length of the grain [21]; and Ray cell walls, which run perpendicular to the grain from the pith to the cambium and are found in both hardwoods and softwoods [23,24].

Ray cells constitute a relatively small portion of the micro-structures in wood, so the main elements that will interact with microwave fields are the tracheids, or vessels or pores, as is the case with hardwoods, which vary in exact shape but generally have an aspect ratio that varies between 2:1 and 4:1. The dielectric properties of wood, at microwave frequencies, are also between 1.2 and 1.9 times higher when the microwave’s electric field is oriented parallel to the wood grain compared with their value when the electric field is oriented perpendicular to the wood grain [7]. The ratio of the dielectric properties along the grain and across the grain may be related to the average aspect ratio of the vessel cells.

### 1.2. Wood Damage

Break down of wood material by termites and decay fungi will affect a significant number of pores within the wood; thus, creating voids in the wood’s micro-structure. These voids will affect the bulk dielectric properties of the wood, which in turn will affect the transmission and mutual coupling response of microwave fields propagating through wood.

Both termites and decay fungi use wood as a food source; therefore, wooden power poles are vulnerable to attack by these biological agents. Currently, pole testing either involves a subjective evaluation based on the acoustics of the pole or sampling from wood shaving cores, which are drilled into the pole itself. Drilling compromises any protection that is afforded by various fungicidal and insecticidal wood treatments; therefore, the development of a simple, objective measurement technique, which does not compromise the pole’s integrity in any way, will be of enormous benefit to the power utility industry. This paper presents the results of an in-situ assessment of wooden power pole stubs using two microwave measurement techniques. Development of a non-destructive device for pole evaluation will provide the industry with an indispensable tool to manage the problem of termite and decay infestation and potentially increase the service life of timber-in-service.

## 2. Materials and Methods

### 2.1. Field Test Sites and Testing Requirement

A field test site was established in the Northern Territory of Australia (12°10′ S, 136°43′ E). This field site was selected due to the major ecotype being eucalypt species. Monterey pine (*Pinus radiata D. Don)* baits (500 × 90 × 45 mm, with grain in the direction of the 500 mm length) were buried just below the soil surface (5–10 cm) in the vicinity of the field site. Each bait station was wetted with water, covered with a black plastic sheet (to remain a moist environment in the bait), then covered with soil (Figure 3). New baits were added every six months to increase and maintain termite pressure at the field site.

Forty power pole stubs of two hardwood species, Spotted Gum (*Corymbia maculate Hook.*) and Blackbutt (*Eucalyptus pilularis Sm.*), where also installed at the Northern Territory testing field site in July 2009 (Figure 4). Spotted Gum and Blackbutt are hardwood species, which are commonly used as power poles in Australia. Each species was represented by 20 poles, half of which were treated with a standard copper chromium arsenic (CCA) wood preservative treatment. Treatment was done at a commercial facility, which commonly treats power poles for power authorities in Australia. The other half of the pole population, from each species, were left as untreated controls. The poles were installed in a randomized design.

### 2.2. Termite Species Found in the Field Site

Although other termite species are known to be present in this field site as subterranean foragers, the major economically important termite species were:*Coptotermes acinaciformis* (Froggatt),*Nasutitermes eucalypti* (Mjoberg)*Microcerotermes* spp.*Heterotermes vagus* (Hill)*Schedorhinotermes intermedius* (Brauer)*Mastotermes darwiniensis* (Froggatt)

### 2.3. A “Look Through” Microwave System

A prototype microwave system (Figure 5) that measures microwave attenuation and phase delay of microwave fields between two antennas that are placed on either side of a wooden structure was developed to detect termites and decay in wood at equilibrium moisture content (EMC) [8]. This microwave system performs a frequency sweep between 2.2 and 2.7 GHz. The transmitting and receiving antennas are identical micro-strip patch antenna arrays, designed to operate at a central frequency of 2.4 GHz with a 14% band width (Figure 6).

This system was tested on fourteen pole stubs at the University of Melbourne’s Burnley campus (37°49′ S, 145°01′ E). All these pole stubs were cut from sound Spotted Gum poles; however, seven were treated with copper chromium arsenic (CCA) wood preservative, while the other seven were untreated. This test provided a base level response for the microwave system.

The poles at the Northern Territory field test site were tested with the prototype system on 21 February 2011, by placing the transmitting and receiving antennas on either side of the pole stub (Figure 6) and capturing the attenuation and phase delay for each pole. The poles were assessed with the antennas at ground level and oriented so that the microwave’s electrical field was parallel to the wood grain. This orientation was chosen based on earlier results from a laboratory experiment that used the same apparatus to successfully detect different levels of decay in wood [8]. The power pole assessment data was analyzed using a two-factor analysis of variance.

The above ground sections of all the poles were visually inspected for evidence of termite and decay damage. A subset of both treated and untreated poles (5 poles from each class) were subsequently lifted out of the ground using an excavator and inspected for below ground termite and decay damage, before being reinstalled for future assessment.

### 2.4. A Radar Based Microwave System

The HB100 (Singapore Technologies Engineering, Singapore, Singapore) microwave motion sensor operates as an X-Band Bi-Static Doppler transceiver module (Figure 7), operating at 10.525 GHz. Its simple design, built-in Dielectric Resonator Oscillator (DRO), built in micro-strip patch antenna arrays and cheapness make the HB100 a potential candidate for determining the properties of wood when the module is placed in direct contact with the surface of wooden power poles. Because the module generates a continuous microwave output, unlike conventional radar, which pulses the microwave output, this system normally responds to Doppler Shift in the reflected field. When the module is placed against a dielectric material, there is strong mutual coupling between the transmitting and receiving antennas on the module. The strength of this mutual coupling depends on the bulk dielectric properties of the space in front of the module. Although this is not the intended purpose of this module, this application represents a significant and innovative repurposing of the HB100 system.

Extensive laboratory experiments (unpublished) demonstrated that this module could be incorporated into a sensor box, along with other environmental parameter sensors, such as temperature and humidity sensors (Figure 8) The response of the radar module sensor boxes was set up so that the data were transmitted via one of the Australian mobile telephone networks from remote sites to a central file server for data storage and further analysis.

Pre-deployment tests were performed on the radar sensor boxes, during which they were calibrated using 5 samples of 200 × 65 × 65 mm *Eucalyptus regnans* (F. Muell.)*,* at equilibrium moisture content (EMC) with the laboratory’s atmospheric conditions, to ensure that all the sensor boxes produced the same output when the radar modules were placed against these wood samples. During these pre-deployment tests, the ability of the sensor boxes was tested to ensure they could detect active termites in timber. Samples of *E. regnans* had a longitudinal hole drilled into their end gain, to allow a known number of termites to be introduced into the timber. The sensor boxes were placed on the samples and tested with no termites in the samples and with 10 worker cast *Coptotermes acinaciformis* termites added to the samples through the longitudinal hole. The output from the radar module was recorded over a period of several minutes to determine the difference in signals when termites were absent or present.

During other pre-deployment experiments, the radar module’s sensor box was set up on a short section of wooden power pole, in the field at the University of Melbourne’s Burnley campus (37°49′ S, 145°01′ E), to confirm the data transmission ability of the system before deployment to the Northern Territory field site. During this pre-deployment test time, the sensor boxes monitored the output during a series of spring storms at the Burnley campus.

In early March 2012, three sensor boxes, based around the HB100 radar modules (Figure 8) were installed at the field site in the Northern Territory. These sensor boxes were wirelessly connected to a server box at the Northern Territory field site, which transmitted the data from the Northern Territory field site to the University of Melbourne’s Burnley site, via the mobile telephone network. The server box was placed on a treated Blackbutt pole stub with a heavy-duty battery buried next to it, for security reasons.

One of the radar sensor boxes was placed on a treated Blackbutt pole stub that was not affected by termite activity. It was used as the control (Sensor NB) during this experiment. A second sensor box was placed on an untreated Blackbutt pole stub, which was slightly affected by termite activity, but appeared not to have active termites within it at the time of testing (Sensor NA). A third sensor box was placed on an untreated Blackbutt pole stub, which had active termites foraging in the pole (Sensor NC). All sensor boxes and the server box were fitted with solar panels, which were placed on top of the pole stubs.

The output from the HB100 radar module was fed into an Arduino analogue to digital converter module (A/D Converter). Measurements were made with the microwave’s electrical field oriented perpendicular to the wood grain. This configuration provided the best response during laboratory experiments using decayed wood samples (Unpublished Data). Raw A/D output data from the three sensors was wirelessly transmitted to the server box, along with date, time, and other environmental data from temperature and humidity sensors. The data from the server box was transferred to the University of Melbourne’s server every 30 min via the mobile telephone network. This data was later downloaded from the server for data visualisation and analysis.

### 2.5. Statistical Analysis

Experimental data was analysed by Analysis of Variance (ANOVA) and post-hoc evaluation of treatment means was performed using Fisher’s Least Significant Difference (LSD) method a the 95% confidence level.

## 3. Results

During the 2011 inspection, it was determined that most of the untreated hardwood poles showed evidence of termite damage to their above ground sections (Figure 9); however, except for one treated pole, which had slight borer damage, the treated poles showed no evidence of deterioration. Inspection of the poles, after extraction from the ground, revealed that all the untreated poles had active *Mastotermes darwiniensis (Froggatt)* and *Coptotermes acinaciformis (Froggatt)* infestation and were severely degraded (Figure 10); however, apart from the one treated pole with slight above ground borer damage, none of the treated poles, which were extracted for inspection, showed any signs of damage or degradation.

### 3.1. A “Look Through” Microwave System

Pretesting of the “look through” system showed that sound, treated Spotted Gum poles significantly attenuated the microwave signals, compared with the sound un-treated poles (Figure 11). Microwave attenuation measurements from the Northern Territory field experiment showed a significant difference between treated and untreated poles; however, there was no difference between timber species (Figure 12). The phase delay data from this look through device was not as clear in differentiating between the treatments (Table 1) as the attenuation data was.

When wood species was not considered in the analysis process, both attenuation and phase delay easily distinguish between the termite affected untreated poles and the sound treated poles (Table 2). Based on this data, the look-through microwave system can distinguish between the termite infested, untreated poles and the sound treated poles.

### 3.2. A Radar Based Microwave System

The pre-deployment testing on the 200 × 65 × 65 mm *Eucalyptus regnans* samples, using the HB100 radar based sensor boxes, demonstrated that these systems could distinguish between timber at laboratory atmosphere EMC with no termites present and timber at laboratory atmosphere EMC with active termites present (Figure 13).

Data from the pre-deployment Burnley field test revealed that the radar sensor boxes were also responding to the moisture content of the short section of wooden power pole. Figure 14 nicely shows the response of the radar module’s output to a unique drying event that was caused by a sudden drop in relative humidity at the Burnley test site during this pre-deployment testing of the system.

Data from the Northern Territory site, using the HB100 radar-based sensor boxes, showed significant differences between the three monitored poles (Figure 15). The high degree of variability in the data from the NC sensor box was due to the detection of foraging termites in this pole stub.

## 4. Discussion

The propagation of microwave fields is highly dependent on the dielectric properties of the wood through which the fields pass. The dielectric properties of most organic materials, including wood, depend on the moisture content of the material (Figure 2) [7,25,26,27,28]. Termites also have a very high moisture content; therefore, their passage through the microwave fields significantly perturb the field propagation, which results in fluctuations of the output from both the ‘look through’ microwave sensor and the radar based microwave system. The additional attenuation in the ‘look through’ system in the sound treated poles, compared with the sound untreated poles (Figure 12), is due to the change in dielectric properties of the wood associated with the addition of the chemical treatment.

After being in the field for approximately 1.5 years, the untreated pole stubs were infested with active *Mastotermes darwiniensis (Froggatt)* and *Coptotermes acinaciformis (Froggatt)*. When tested, the attenuation in the treated poles was higher (see Figure 12) than in the pre-test case (see Figure 11). This was probably due to higher moisture content in the wood due to the tropical location of the Northern Territory field experiment. The field site data also revealed that, instead of the attenuation in the untreated poles being significantly less than in the treated poles, as in the case of the pre-test, the attenuation was significantly higher than in the treated poles. This is attributable to the very high moisture content and damage caused to the poles by termite and decay infestation in these poles in the field.

Testing the 40 poles, at the experimental field site, took one hour and four minutes to complete, which included shifting the prototype equipment from one pole to the next. This equates to a testing time of 1.6 min per pole, which is significantly faster than conventional pole testing techniques, including: visual inspection; the hammer test, where the resonant sound of the pole is evaluated after striking it with a heavy hammer; and bore testing, where holes are drilled into the pole to evaluate the condition of the wood shavings that are extracted during this invasive procedure.

The system, based on the HB100 radar module, was also effective at detecting termite activity in both laboratory (Figure 13) and in-situ conditions (Figure 15). The system can also detect moisture in the timber (Figure 14). It is interesting to note the time delay between the drop in humidity and the corresponding response in the microwave data. This delay is consistent with the time delay required for moisture to diffuse out of the timber in response to the change in the humidity of the air surrounding the timber [29].

It has already been demonstrated that the ‘look through’ system can distinguish between sound timber (both hardwood and softwood) and timber that has been degraded by decay [8]. Although, not yet published, the HB100 system can also distinguish between sound timber (both hardwood and softwood) and timber that has been degraded by decay. The generally higher mean output from the NC sensor at the field site (Figure 15) was also indicative of higher moisture content in the wooden pole, due to the termite infestation in the pole.

The data presented in (Figure 15) span three days. It is interesting to note the cyclic undulations in all three data sets. This is also possibly due to the diurnal variation in humidity in this tropical environment, as was revealed in the data shown in Figure 14. Upon exposure to the atmosphere, the moisture levels in freshly harvested timber become thermodynamically unstable and the cell structures dry until they reach equilibrium with the local atmosphere. This state is called equilibrium moisture content (EMC) [21], with moisture content being measured on a dry weight basis [30]. EMC is related to the humidity and temperature of the local atmospheric conditions [31]. Based on equations given by Simpson [31], the EMC of undegraded wood may fluctuate between 3% (under desert conditions) and 29% (under wet tropical conditions). It is also important to note that although all of the data traces change due to these fluctuations in humidity and temperature from day to night (Figure 15), the fluctuations in the degraded poles are much higher than in the sound pole.

It is also important to note the rapid undulations in the ‘Active Termites (Sensor Box NC)’ data, in Figure 15, compared with the other two data traces. These rapid undulations are typical of active termite movement within the microwave fields and is a clear indicator that the degradation of the timber is due to active termite attack rather than general degradation or decay.

One advantage of the radar-based system is that, unlike the ‘look through’ system, it does not require access to both sides of the wood sample to be effective.

This experiment has shown the potential for microwave-based sensor systems for monitoring the status of power poles in-situ. These technologies could be incorporated into a simple hand held device for rapid assessment by a linesman or they could be permanently attached to poles in the field as part of an Internet of Things (IoT) enabled network of sensors to continually monitor the status of poles in service.

## 5. Conclusions

Exposure to tropical conditions resulted in termite infestations in all untreated hardwood poles; however, while some of the untreated poles had active termite infestations, others only showed evidence of termite damage with no termites being observed at the time of inspection and testing. There was no evidence of termite attack in any of the treated poles. Both microwave prototypes were able to distinguish between the termite infested, untreated poles and the sound treated poles. The average time taken to test a single pole using the “look though” prototype system was 1.6 min, which is significantly less time than is needed for existing pole testing methods. The radar module sensor boxes were able to provide continuous monitoring of bio-deterioration and termite activity in the poles.

## Figures and Tables

**Figure 1 insects-11-00568-f001:**
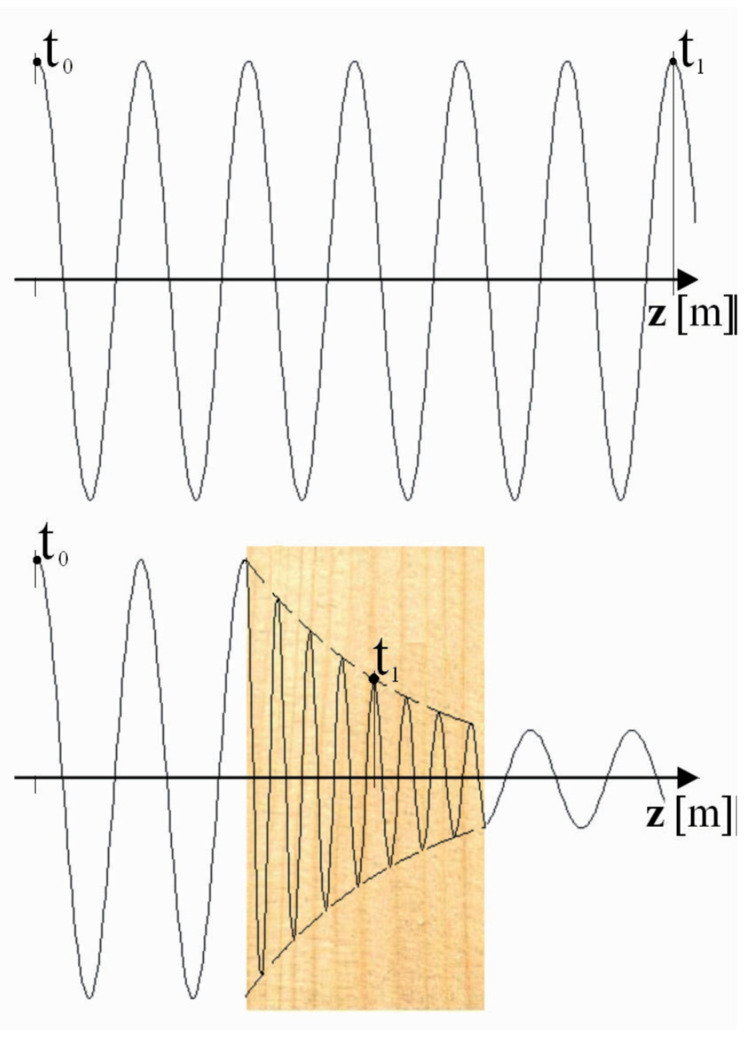
Propagation of microwave energy through air or open space (**top**) and through wood (**bottom**) [18].

**Figure 2 insects-11-00568-f002:**
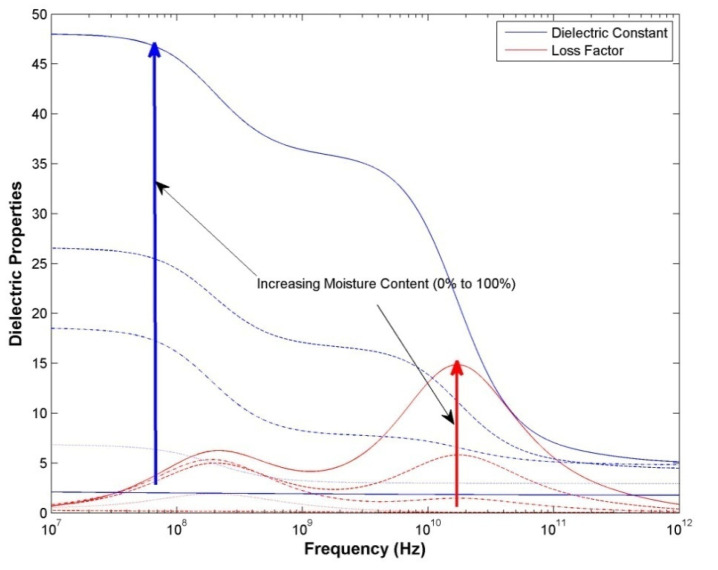
Dielectric properties of wood with the electric field parallel to the grain (density of 600 kg m^−3^) as a function of frequency and moisture content, based on data from Torgovnikov [7]

**Figure 3 insects-11-00568-f003:**
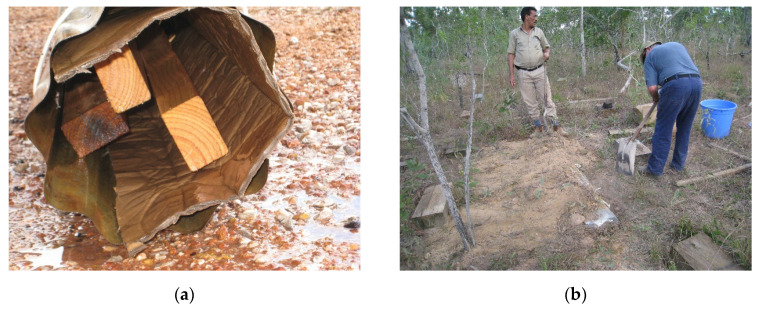
Termite bait station (**a**) during preparation and (**b**) being installed by the authors.

**Figure 4 insects-11-00568-f004:**
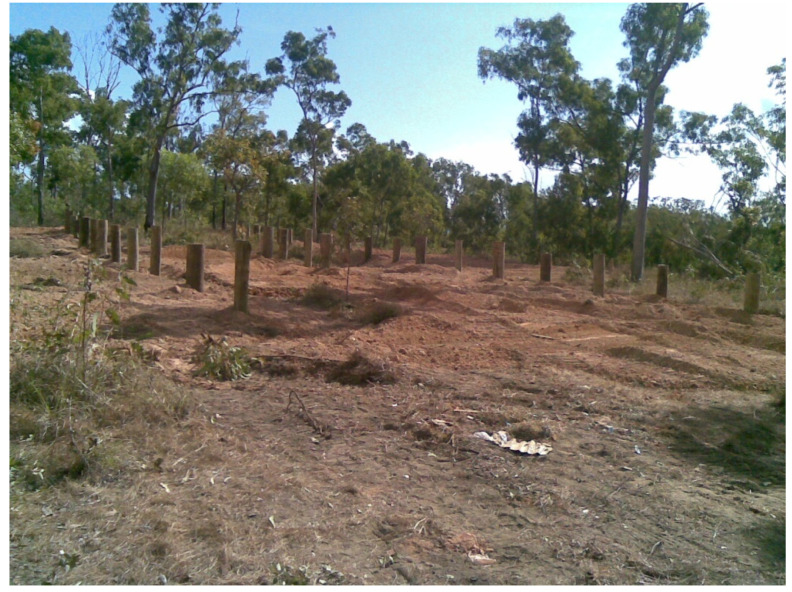
Power pole stubs installed at experimental site.

**Figure 5 insects-11-00568-f005:**
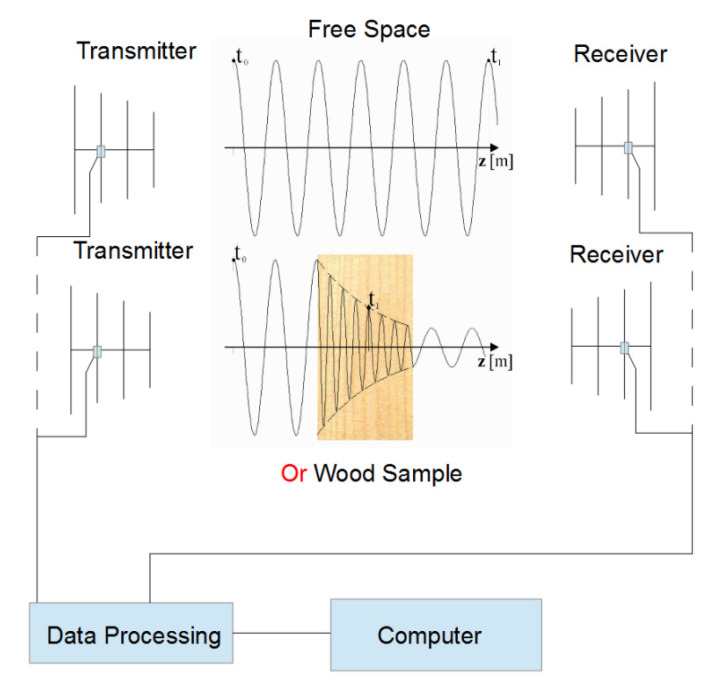
Schematic diagram of the sensor system for the microwave detection of termites and decay in wood with the wave pattern for open space/air shown at the top and the wave pattern for passing through wood shown below.

**Figure 6 insects-11-00568-f006:**
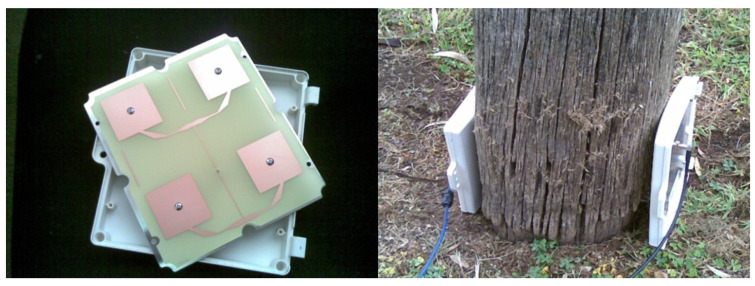
Microstrip antenna array (**left**) showing detail of the array and (**right**) showing the antennas in place for microwave testing of a power pole.

**Figure 7 insects-11-00568-f007:**
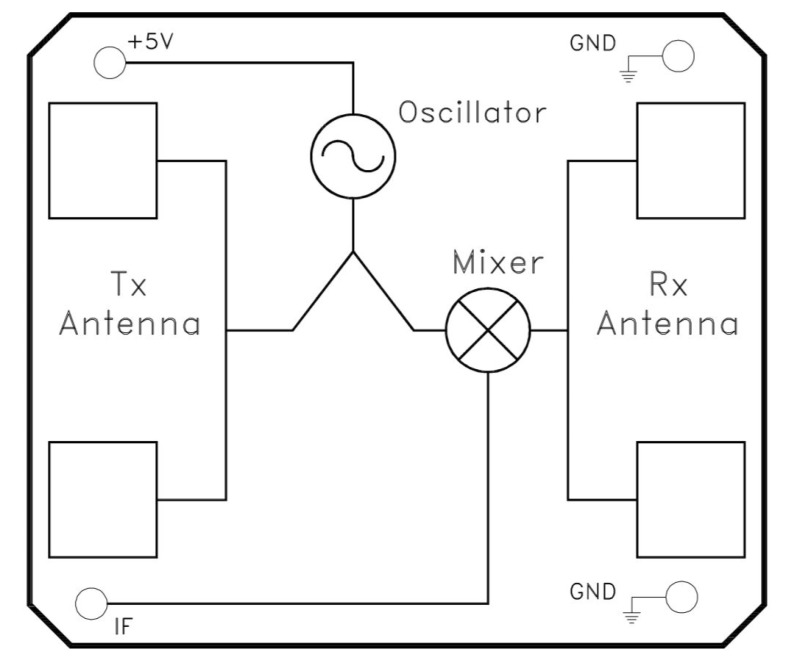
Block diagram for HB100 radar module.

**Figure 8 insects-11-00568-f008:**
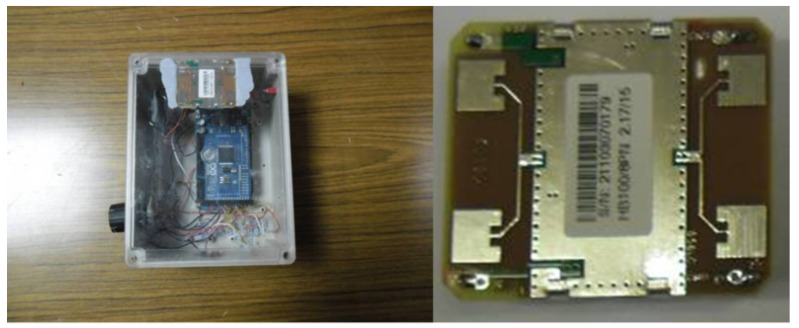
(**Left**) Sensor box with the HB100 radar module (**right**).

**Figure 9 insects-11-00568-f009:**
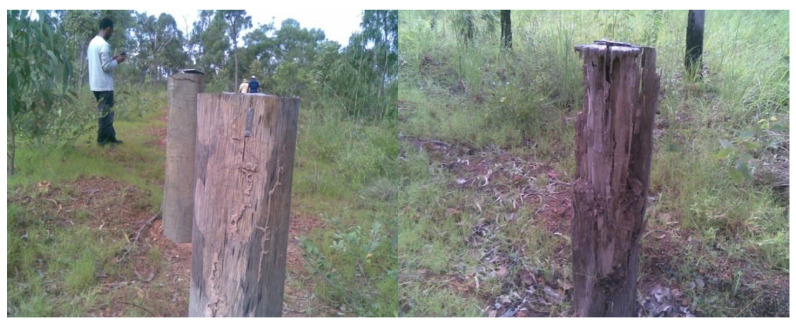
Untreated poles showing varying degrees of termite attack.

**Figure 10 insects-11-00568-f010:**
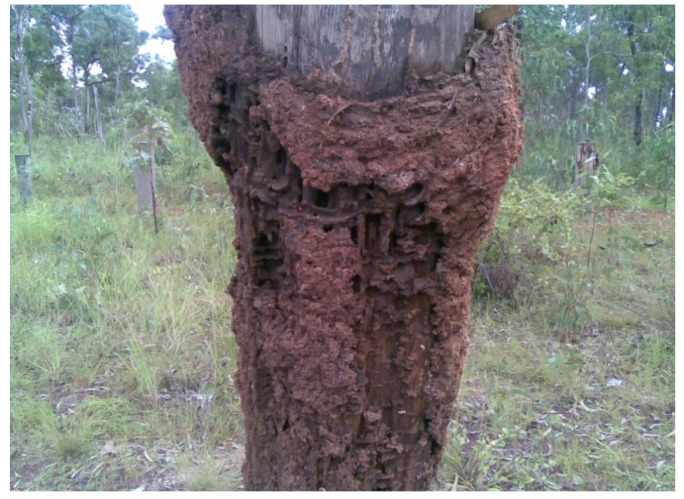
Extracted untreated pole showing active *Mastotermes darwiniensis* infestation.

**Figure 11 insects-11-00568-f011:**
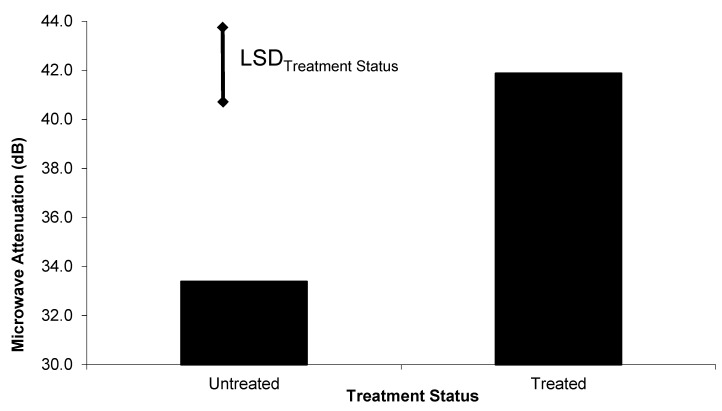
Mean microwave attenuation (dB) in sound Spotted Gum pole stubs as a function of timber treatment status.

**Figure 12 insects-11-00568-f012:**
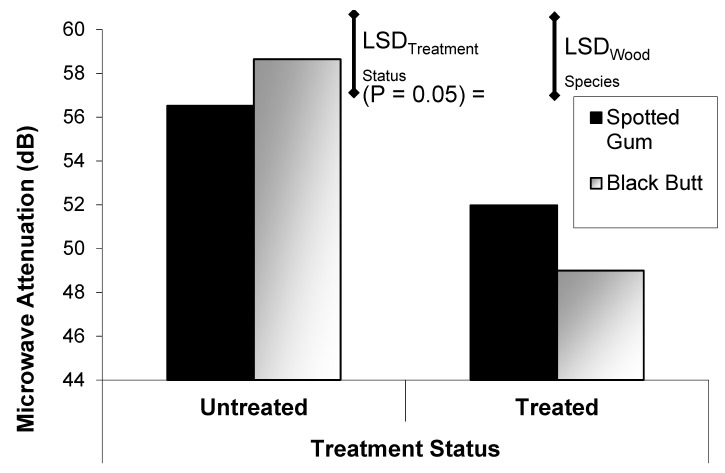
Mean microwave attenuation (dB) in pole at the Northern Territory field site as a function of timber treatment status and species.

**Figure 13 insects-11-00568-f013:**
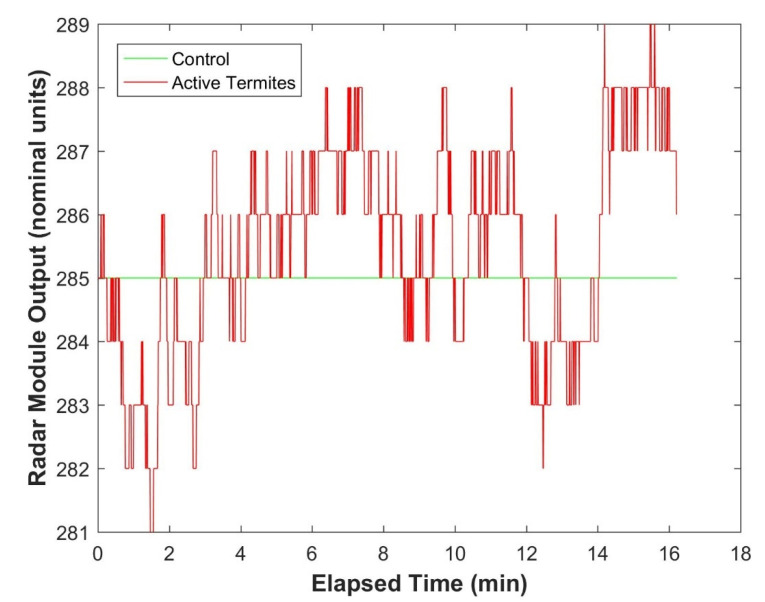
Comparison of radar module output for samples of timber at laboratory atmosphere equilibrium moisture content (EMC) with and without (Control) active termites present.

**Figure 14 insects-11-00568-f014:**
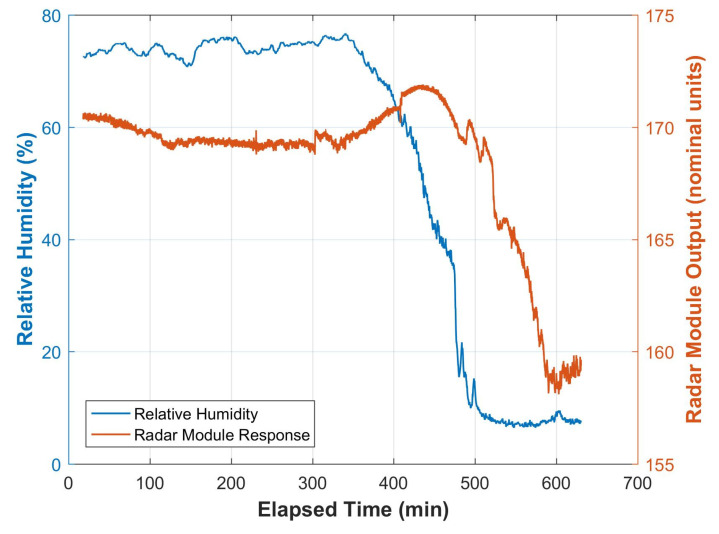
Output from a HB100 radar module, applied to a wooden sample block (30 mm by 30 mm by 200 mm), in response to changing external humidity.

**Figure 15 insects-11-00568-f015:**
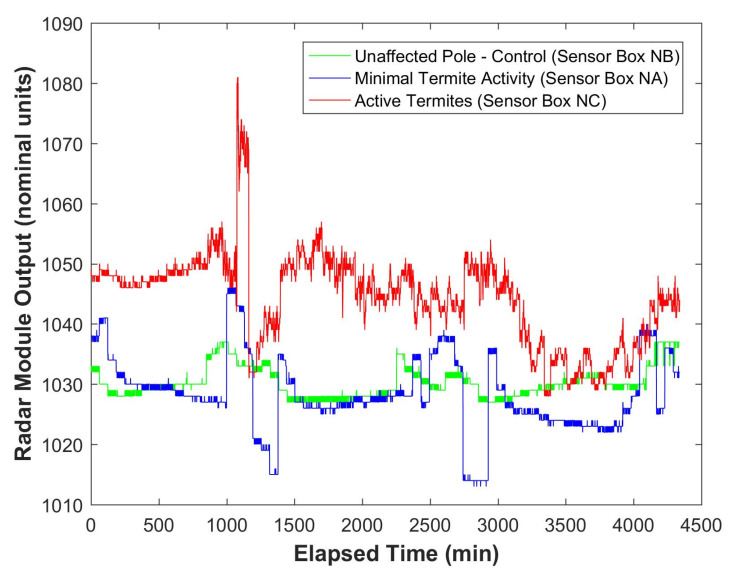
Radar response for three poles installed in the field in the Northern Territory.

**Table 1 insects-11-00568-t001:** Mean microwave phase delay (picoseconds).

Species	Treatment Status	
	Untreated	Treated
Spotted Gum	22533.7 ^a^	21762.4 ^a b^
Blackbutt	22499.4 ^a^	21232.4 ^b^
LSD (*p* = 0.05)		986.7

Note: Means with different superscripts are significantly different from one another.

**Table 2 insects-11-00568-t002:** Mean microwave measurement data for treated and untreated pole condition only.

Microwave Measurement	Untreated	Treated	LSD (*p* = 0.05)
Attenuation (dB)	57.6 ^a^	50.5 ^b^	2.1
Phase delay (Picoseconds)	22516.5 ^a^	21497.4 ^b^	697.7

Note: Means in each row with different superscripts are significantly different to one another.

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
