# Peer review of "An In-Situ Assessment of Wood-in-Service Using Microwave Technologies, with a Focus on Assessing Hardwood Power Poles"

_insects, 2020, doi:10.3390/insects11090568_

Round 1

Reviewer 1 Report

The study is sound and easy to follow. Here are a few considerations:

  • line 57-58: This phase is quite lost. It should be expanded to better feet in the context
  • line 99-105: A figure about the bait station. It will improve the manuscript
  • line 106-107: Why these species were choose? Are they the most used for wooden poles in Australia? 
  • line 109: How were the specimens treated? Were they treated in the lab or bought in the market? Please, add this information
  • line 117-122: Scientific names should always be italicized
  • line 131-133: This information is repeat. See lines 108-111
  • line 150-151: Did the authors use a visual rating system do conducted this evaluation? Please, add it to the text

Author Response

Response to Reviewer 1

Reviewer Comment

Response

This phase is quite lost. It should be expanded to better feet in the context

“Hardwood vessels serve as the pipelines within the trunk, transporting sap within the living tree. They extend through the trunk and branches, providing longitudinal connections along the length of the timber. Resin channels are resin filled cavities in the wood structure. Both vessels and resin channels affect the physical and electromagnetic properties of timber.” Has been added to the text to help expand the comment on vessels and channels.

A figure about the bait station. It will improve the manuscript

Figure 4 added to the manuscript to illustrate the form and installation of the bait stations. Other figure numbers updated to accommodate the new Figure 4.

Why these species were choose? Are they the most used for wooden poles in Australia?

Yes, these species are commonly used for power poles. The text “Spotted Gum and Blackbutt are hardwood species, which are commonly used as power poles in Australia.” Has been added to the text.

How were the specimens treated? Were they treated in the lab or bought in the market? Please, add this information

As already stated in the existing text, the poles were treated with “standard copper chromium arsenic (CCA) wood preservative treatment”. The text “Treatment was done at a commercial facility, which commonly treats power poles for power authorities in Australia.” Has been added.

Scientific names should always be italicized

This has been updated in the text

This information is repeat. See lines 108-111

In fact, this is not repeated information. The poles referred to in lines 108-111 were installed in the Northern Territory field site. The poles referred to in lines 131-133 were at our Burnley campus in Melbourne (at the opposite end of the country) and used to calibrate and test the technology before deployment at the Northern Territory site.

Did the authors use a visual rating system do conducted this evaluation? Please, add it to the text

No ‘standardised’ grading system was used on this occasion. Untreated poles were visually degraded and infested with termites while treated poles were unaffected by termites and visually very sound in appearance.

The following text has been added in the beginning of the Results section to help address the reviewer’s concerns “Inspection of the poles, after extraction from the ground, revealed that all the untreated poles had active Mastotermes darwiniensis (Froggatt) and Coptotermes acinaciformis (Froggatt) infestation and were severely degraded (Figure 11); however, apart from the one treated pole with slight above ground borer damage, none of the treated poles, which were extracted for inspection, showed any signs of damage or degradation.”

Reviewer 2 Report

This is a very interesting study and the results would be of great interest to electrical utilities. I did have a difficult time following the discussion on the effects of wood decay. The title indicates that you are testing for both, but the discussion is heavily weighted towards termites. Please add some discussion and potentially results that demonstrate the capabilities of this method to detect decay as well as termites or indicate that these types of biodeterioration cannot be distinquished from each other using this methods. The ideal scenario would be a termite free control where wood destroying insects are excluded from the sample to isolate the effects of wood decay fungi. This shouldn't detract from the applied value of this work, but only strengthen its basic scientific value. 

Author Response

Response to Reviewer 2

Reviewer Comment

Response

This is a very interesting study and the results would be of great interest to electrical utilities. I did have a difficult time following the discussion on the effects of wood decay. The title indicates that you are testing for both, but the discussion is heavily weighted towards termites. Please add some discussion and potentially results that demonstrate the capabilities of this method to detect decay as well as termites or indicate that these types of biodeterioration cannot be distinquished from each other using this methods. The ideal scenario would be a termite free control where wood destroying insects are excluded from the sample to isolate the effects of wood decay fungi. This shouldn't detract from the applied value of this work, but only strengthen its basic scientific value.

The following text has been added to the Discussion section of the paper: “It has already been demonstrated that the ‘look through’ system can distinguish between sound timber (both hardwood and softwood) and timber that has been degraded by decay [8]. Although, not yet published, the HB100 system can distinguish between sound timber (both hardwood and softwood) and timber that has been degraded by decay.”

Further, the following text has been added to the same paragraph in the discussion: “It is also important to note the rapid undulations in the ‘Active Termites (Sensor Box NC)’ data, in Figure 16, compared with the other two data traces. These rapid undulations are typical of active termite movement within the microwave fields and is a clear indicator that the degradation of the timber is due to active termite attack rather than general degradation or decay.”

Reference from the new section of the Discussion

8.             G. Brodie, M. V. Jacob and B. M. Ahmed. Microwave characterisation and detection of wood decay. World Journal of Engineering 2012, 9, 265-272

Reviewer 3 Report

Overall

This manuscript focuses on the practical application of 2 microwave-based technologies for assessing biological deterioration of hardwood power poles. The experiments are well designed and the results are very interesting. However, the authors should add information/discussion according to the following comments.

Specific comments

Major points:

#1 In Introduction, the authors should refer the microwave termite detection apparatus, Termatrack, because it is used for detecting termite infestation world-wide. What’s technological difference between Termatrack and the authors’ apparatuses?

#2 L55-80: In Figure 3, the left drawing is strange. Tracheids are in softwood and vessels are in hardwood. The right drawing shows nothing. I recommend to delete Figure 3. How about fibers, the major element of hardwood? Can pores in wood interact with microwave? Anyway, I can’t catch the points what the authors want to say in this part. What relationship between wood microstructure and termite/decay infestation in terms of microwave attenuation? The authors should make a clear explanation with data. In my opinion, the macrostructure, such as heartwood/sapwood ratio, earlywood/latewood ratio and interval, etc. is more effective on termite/decay infestation, leading to microwave attenuation.

#3 L272 - : We know that microwave is easily absorbed by water and is hard to be applied to the high MC materials such as wet wood. The authors should make a clear discussion on how to separate 2 types of poles: only high MC and high MC + biological deterioration in the practical situation.

Minor points

#4: L115-122: The authors’ description “3. Microcerotermes spp.” means multiple species of Microcerotermes, which shows total more than 6 termite species, not six species.

#5: L155: Need information of HB100: company name etc.

#6: The authors should add information on statistical analyses after L212.

#7: The authors use both “Melbourne University” and “the University of Melbourne”. Please check it.

#8: What are “conventional pole testing techniques” in L292?

Author Response

Response to Reviewer 3

Reviewer Comment

Response

In Introduction, the authors should refer the microwave termite detection apparatus, Termatrack, because it is used for detecting termite infestation world-wide. What’s technological difference between Termatrack and the authors’ apparatuses?

The following text has been added to the Introduction: “The Termatrac® system is a commercial example of using microwave energy to detect termite activity in timber. It is a radar based system, which responds to movement [12]. The maximum reliable wood depth for detecting termites was 5 cm [13], which is limiting for evaluation of larger wooden structures, such as power poles. Evaluation of the Termatrac® system revealed that operator skill was critical in establishing its efficacy in termite detection. For example, Zahid, et al. [12] found that one of their test operators appeared to be better than the other with a sensitivity of 66% compared to 45% for the second operator. While this could be overcome with training, their study found that the efficacy of the Termatrac® system was lower than several other termite detection systems. Another limitation of the Termatrac® system is that it is expensive and not convenient for permanent installation.”

References

12.           I. Zahid, C. Grgurinovic, T. Zaman, R. De Keyzer and L. Cayzer. Assessment of technologies and dogs for detecting insect pests in timber and forest products. Scandinavian Journal of Forest Research 2012, 27, 492-502

13.           S. Taravati. Evaluation of Low-Energy Microwaves Technology (Termatrac) for Detecting Western Drywood Termite in a Simulated Drywall System. Journal of Economic Entomology 2018, 111, 1323

In Figure 3, the left drawing is strange. Tracheids are in softwood and vessels are in hardwood. The right drawing shows nothing. I recommend to delete Figure 3. How about fibers, the major element of hardwood? Can pores in wood interact with microwave? Anyway, I can’t catch the points what the authors want to say in this part. What relationship between wood microstructure and termite/decay infestation in terms of microwave attenuation? The authors should make a clear explanation with data. In my opinion, the macrostructure, such as heartwood/sapwood ratio,

© 1996-2020 MDPI (Basel, Switzerland) unless otherwise stated

earlywood/latewood ratio and interval, etc. is more effective on termite/decay infestation, leading to microwave attenuation.

Figure 3 has been removed and figure numbers have been updated appropriately.

We know that microwave is easily absorbed by water and is hard to be applied to the high MC materials such as wet wood. The authors should make a clear discussion on how to separate 2 types of poles: only high MC and high MC + biological deterioration in the practical situation.

The following text has been added to the Discussion section: “Upon exposure to the atmosphere, the moisture levels in freshly harvested timber become thermodynamically unstable and the cell structures dry until they reach equilibrium with the local atmosphere. This state is called equilibrium moisture content (EMC) [21], with moisture content being measured on a dry weight basis [30]. EMC is related to the humidity and temperature of the local atmospheric conditions [31]. Based on equations given by Simpson [31], the EMC of undegraded wood may fluctuate between 3 % (under desert conditions) and 29 % (under wet tropical conditions). It is also important to note that although all of the data traces change due to these fluctuations in humidity and temperature from day to night (Figure 15), the fluctuations in the degraded poles are much higher than in the sound pole.”

The authors’ description “3. Microcerotermes spp.” means multiple species of Microcerotermes, which shows total more than 6 termite species, not six species

The reference to ‘six’ species has been deleted.

Need information of HB100: company name etc.

“(Singapore Technologies Engineering, Singapore)” Added to the text

The authors should add information on statistical analyses after L212.

The following text was added to the text:

“2.5. Statistical Analysis

Experimental data was analysed by Analysis of Variance (ANOVA) and post-hoc evaluation of treatment means was performed using Fisher's LSD method to the 95 % confidence level.”

The authors use both “Melbourne University” and “the University of Melbourne”. Please check it.

Thank you. These have been corrected.

What are “conventional pole testing techniques” in L292?

The following text was added: “including: visual inspection; the hammer test, where the resonant sound of the pole is evaluated after striking it with a heavy hammer; and bore testing, where holes are drilled into the pole to evaluate the condition of the wood shavings that are extracted during this invasive procedure.”

Round 2

Reviewer 3 Report

Thank you for revising the manuscript according to my comments. I have gone through the revised manuscript and would like to accept all the revisions.